# P-ALIGN: SELF-ALIGNMENT IN PHYSICAL DYNAMICAL SYSTEM MODELING

## ABSTRACT

Deep learning has emerged as the new paradigm in modeling complex physical dynamical systems. Nevertheless, data-driven methods learn patterns by optimizing statistical metrics, tend to overlook the adherence to physical laws. Previous work have attempted to incorporate physical constraints into neural networks, but they often face limitations due to lack of flexibility or optimization challenges. In this paper, we propose a novel framework, *Physics-aware Self-Alignment* (P-ALIGN), to enhance the physical consistency of dynamical systems modeling. P-ALIGN enables dynamical system models to provides physics-aware rewards, which makes self-alignment of dynamical system models possible. Comprehensive experiments show that P-ALIGN not only gave an average statistical skill score boost of more than 32% for ten backbones on five datasets, but also significantly enhances physics-aware metrics. All of our source codes will be released via GitHub.

## 1 INTRODUCTION

Dynamical systems provides a mathematical framework for analyzing how systems evolve over time, which is particularly important in fields such as fluid mechanics, climatology and meteorology. It describes the temporal evolution of a system's state using differential equations for continuous systems or difference equations for discrete systems (Birkhoff, 1927; Anosov et al., 1988; Meiss, 2007; Galor, 2007). However, solving these equations analytically is often not feasible for complex systems, leading to a reliance on numerical methods (Stuart & Humphries, 1998; Dellnitz & Junge, 2002; Guckenheimer, 2002; Hubbard & West, 2012). While numerical approaches such as finite difference methods, finite element methods or Runge-Kutta methods can provide approximate solutions (Lisitsa et al., 2012; Thomas, 2013; De La Cruz et al., 2013), they tend to be computationally expensive, especially for high-dimensional systems or long time spans (Houska et al., 2012; Benner et al., 2015; Yu & Wang, 2024).

Data-driven approaches to dynamical systems modeling have gained significant attention as a way to overcome some of the limitations of traditional numerical methods (Pfaff et al., 2021; Gao et al., 2022b; Pathak et al., 2022; Bi et al., 2023; Wu et al., 2024a). These approaches leverage large datasets and deep learning to model the underlying dynamics directly from observed data, bypassing the need for explicit analytical forms of the governing equations (Yu & Wang, 2024). By capturing complex behaviors through data, these methods offer a promising alternative for modeling high-dimensional, nonlinear, or chaotic systems where traditional approaches struggle (Noé et al., 2020; Wang et al., 2020; Kochkov et al., 2021). Nevertheless, data-driven methods often build models by optimizing statistical metrics, which can lack the physical consistency that traditional methods based on first-principles offer (Han et al., 2020; Karniadakis et al., 2021; Pathak et al., 2022). Without explicitly incorporating physical constraints, deep learning models may produce predictions that, while statistically accurate, are physically implausible or violate fundamental physical laws (Pathak et al., 2022; Bi et al., 2023; Wu et al., 2024b). This limitation is especially problematic when extrapolating beyond the range of the training data, where the model may generate behavior that contradicts well-established physical principles (Willard et al., 2020; Wang et al., 2021).

Recent research explored various methods to introduce physical constraints to the data-driven approach to enhance the physical consistency of the prediction. Some methods (Raissi et al., 2019; Li et al., 2021; Hansen et al., 2023) incorporate physical equations as part of the training process to ensure adherence to physical laws, but optimization challenges, especially with complex constraints,

often lead to suboptimal results (Krishnapriyan et al., 2021). Some other methods integrates physics-inspired components into neural networks (Greydanus et al., 2019; Cranmer et al., 2020), but this approach requires clearly defining physical rules and developing custom architectures, limiting its flexibility to different task and backbone network. More recent models (Gao et al., 2023) employ a physics-informed energy function to guide the sampling process, but the need for an additional network to align the physical constraints increases the overall complexity of training.

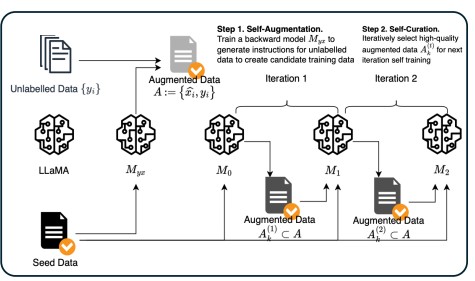 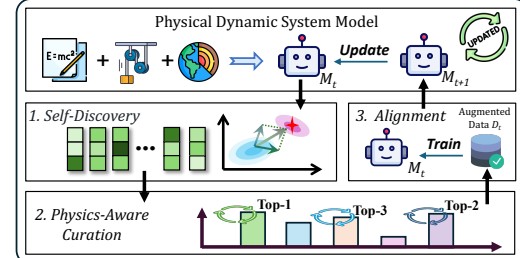

(a). Self-Alignment in Large Language Models       (b). Self-Alignment in Dynamical System Modeling

Figure 1: The figure shows two self-improvement frameworks: Figure (a) illustrates the self-alignment of large language models (Li et al., 2023), while Figure (b) presents the self-alignment in physical dynamic system modeling, which draws inspiration from the LLM's self-alignment in Figure (a). Both achieve self-improvement and capability enhancement through iterative self-discovery, self-filtering, and self-updating.

In this paper, inspired by the self-alignment in large language models, as shown in Figure 1, we introduce *Physics-aware Self-Alignment* (P-ALIGN), a novel self-alignment framework designed to enhance the physical consistency of dynamical system models. P-ALIGN aims to enable dynamical system models with two key capabilities simultaneously: **(1) Prediction**: accurately forecasting future states based on the current state. **(2) Curation**: generating and evaluating multiple potential future states with high physical consistency to expand and improve the training dataset. With these capabilities, dynamic system models can train themselves iteratively. Our *theoretical analysis* demonstrates that P-ALIGN can improve the performance of the model by reducing the upper bound of the generalization error of the model. Our *experiments* demonstrate that P-ALIGN boost performance in a wide range of dynamic system modeling tasks.

## 2 RELATED WORK

***Data-Driven Dynamical System Modeling***: In the scientific computing field, data-driven physical dynamical system modeling has become an innovative tool. It provides accuracy and insight for solving complex problems in dynamic systems (Reichstein et al., 2019; Wang et al., 2023). This approach allows researchers to deeply understand and model natural phenomena (Long et al., 2018; Chen et al., 2018; Kiani Shahvandi et al., 2022; Höge et al., 2022; Mehta et al., 2021). Applications range from the long-term effects of climate change to the simulation of high-speed fluid dynamics (Pathak et al., 2022; Bi et al., 2022; Li et al., 2020; Xiong et al., 2023). For example, FNO demonstrates excellent performance in processing complex partial differential equations (Li et al., 2020); LSM is effective in data compression and feature extraction (Wu et al., 2023); PINN combines deep learning and physics principles to effectively solve the challenges of traditional numerical methods (Karniadakis et al., 2021); and DeepONet learns universal operators for complex systems and effectively predicts system behaviors (Lu et al., 2021), among other fields.

***Self-Alignment***: Self-Alignment derived from research on large language models, which focuses on enabling models to autonomously improve by generating and evaluating their own data, thereby reducing the need for external supervision (Li et al., 2023; Guo et al., 2024; Liang et al., 2024a). The standard approach (Xu et al., 2023; Sun et al., 2024b; Wang et al., 2024a) involves writing a set of prompts based on specific principles, guiding the model to assess the quality of its generated output (Sun et al., 2024b; Lu et al., 2024), and then using these assessments to fine-tune the model itself. Other methods have attempted to train reward models for judgment (Gulcehre et al., 2023; Sun et al., 2024a) or leverage instruction-tuned models to aid in generating synthetic datasets (Yuan et al.,

Figure 2: The P-ALIGN framework optimizes dynamic system models and aligns them with physical principles through four steps: **Self-Discovery, Physics-Aware-Curation, Data Augmentation**, and **Alignment**. This process enhances the model's physical consistency and predictive accuracy.

2024; Liang et al., 2024b). However, dynamic system models lack the capacity to generate multiple responses and evaluate them, making it challenging to generalize the self-alignment approach.

## 3 PRELIMINARIES

**Dynamical System Modeling:** A typical dynamical system contains multiple variables with spatio-temporal relationships (Anosov et al., 1988; Meiss, 2007; Yu & Wang, 2024). It is described by equations associated with unknown functions and their derivatives as a $k$-th order system of partial differential equation:

$$\mathcal{F}(D^k x(s), D^{k-1} x(s), \ldots, Dx(s), x(s), s) = 0 \tag{1}$$

where $s \in$ the domain $S$ and $x$ means the state of the system. When the governing equation $\mathcal{F}$ is known, we can solve them by some numerical schemes with high-cost computation. Even more often $\mathcal{F}$ is unknown, which makes the numerical schemes completely infeasible.

Data-driven modeling of dynamical systems encourages end-to-end prediction, thereby skipping expensive numerical integration. Specifically, dynamical system models learn to build a probabilistic model $\mathcal{P}(\mathcal{Y}|\mathcal{X};\theta)$, which maps a sequence of past values to future values:

$$\mathcal{M} : \{\mathcal{X}_1, \mathcal{X}_2, \ldots, \mathcal{X}_T\} \to \{\mathcal{Y}_1, \mathcal{Y}_2, \ldots, \mathcal{Y}_T\}, \tag{2}$$

where $\mathcal{Y}_t = \mathcal{X}_{t+\triangle t}$. $\mathcal{X}_t$ is the input features and the $\mathcal{Y}_t$ is the outputs, $T$ stands for forecasting horizon and $\triangle t$ is the time lag.

## 4 METHODOLOGY

### 4.1 FRAMEWORK OVERVIEW

As shown in Figure 2, P-ALIGN explores the latent space of dynamical system models and produces augmented training data with more physical consistency, followed by self-alignment across multiple iterations. Specifically, P-ALIGN performed the following three steps in one iteration: **(1) Self-Discovery**, which identify approximate representative samples from the continuous space in which the current hidden state resides. **(2) Physics-Aware-Curation**, which find new samples with more physical consistency.**(3) Data Augmentation** add the augmented samples to training data. **(4) Alignment**, which use the argument data to train models themselves.

Through these four steps, P-ALIGN enables dynamical system models to achieve self-alignment. Indeed, we construct a reward strategy here: *Self-Discovery* generates candidate responses, which

are then evaluated by *Physics-Aware-Curation* to ensure the physical consistency of the data. Our self-alignment is accomplished by iterative training. For each iteration, a new model $\mathcal{M}_t$ is produced, where $\mathcal{M}_t$ is trained on the augmented data $\mathcal{D}_{t-1}$, generated by the previous model $\mathcal{M}_{t-1}$.

## 4.2 PHYSICS-AWARE SELF-CURATION

***Encoder:*** We first model the features of dynamical system into a embedding space with an encoder, which computes latent vectors from raw observation data in historical inputs. Specifically. the encoder $E_\phi$ takes high-dimensional features of physical systems $\mathcal{X}_t \in \mathbb{R}^{C \times H \times W}$ and maps it to the latent representation $\mathcal{Z}_t \in \mathbb{R}^{n \times D}$ through a series of transformations, where $n$ is the number of tokens and $D$ is the dimension of each token. This transformation process can be described as:

$$\mathcal{Z}_t = E_\phi(\mathcal{X}_t) = \{z_{t_1}, z_{t_2}, \ldots, z_{t_n}\} \tag{3}$$

where each token $z_{t_n} \in \mathbb{R}^D$ corresponds to $D$-dimensional state of local features. As a general method, P-ALIGN can employ any popular backbone networks as the encoder $E_\phi$, such as vision transformer (Dosovitskiy et al., 2021), Earthfarseer (Wu et al., 2024a), SimVP (Tan et al., 2022), FNO (Li et al., 2020), or CNO (Raonic et al., 2024).

***Self-Discovery:*** After mapping the high-dimensional feature $\mathcal{X}_t$ to latent representation via the encoder $E\phi$, our goal is to identify representative sample that approximate $\mathcal{Z}_t$. However, discovery within a high-dimensional continuous space is challenging due to its density. In P-ALIGN, we proposed *Self-Discovery* mechanism. anchor the representative points in a low-dimensional embedded space to represent local features of current state, inspired by latent space traversal (Chalumeau et al., 2023; Adolphs et al., 2022) and vector quantization (Van Den Oord et al., 2017). Specifically, we assumes that a $d$–dimensional ($d < D$) embedding space of dynamical system models can be divided into $N$ sub-regions, which can be formulated as $\mathcal{E} = \{\mathcal{E}_1, \mathcal{E}_2, \ldots, \mathcal{E}_N\}$. For each sub-region $\mathcal{E}_K$, there exists a feature vector $e_k \in \mathbb{R}^d$ capable of representing an approximation of vectors within $\mathcal{E}_K$, which can be defined as:

$$\forall x \in \mathcal{E}_k, \quad e_k = f(x), \quad \text{where} \quad k = \arg\min_j \|x - e_j\|^2 \tag{4}$$

We refer to these representative vectors as anchors. Consequently, the continuous embedding space $\mathcal{E}$ can be approximated by a discrete set of anchors $\{e_k\}$:

$$\mathcal{E} \approx \{e_k\}_{k=1}^N \tag{5}$$

where $N$ is the number of sub-regions. In this formulation, the discovery of continuous space is transformed into the task of discrete vector search. We then align the current state $\mathcal{Z}_t$ to the $d$-dimensional embedding space:

$$\mathcal{Z}'_t = \sigma\left(W \cdot \mathcal{Z}_t + b\right) = \{z'_{t_1}, z'_{t_2}, \ldots, z'_{t_n}\} \tag{6}$$

where $\mathcal{Z}'_t \in \mathbb{R}^{n \times d}$ and $z'_{t_i} \in \mathbb{R}^d$. $W$ and $b$ are the projection matrix and bias, $\sigma$ is the activation function. Then we use the set of anchors $\{e_k\}$ to approximate it:

$$f(z'_{t_i}) = e_k, \quad \text{where} \quad k = \arg\min_j \|z'_{t_i} - e_j\|^2 \tag{7}$$

$$\mathcal{Z}'_t \approx \{e_{k_1}, e_{k_2}, \ldots, e_{k_n}\} \tag{8}$$

For the $d$-dimensional representation $z'_{t_i}$ of each local feature, we select the anchor $e_k$ with the smallest Euclidean distance to approximate it during training and inference, as shown in Equation 6. However, we do not restrict this selection to only the nearest $e_k$ when updating the train dataset; instead, we expand the search space to yield a more diverse set of potential samples. This extension is based on our observation that the embedding space cannot be accurately modeled, resulting in similar statistical metrics for the *top-K* nearest anchors. The search space expansion can be defined as follows:

$$f(z'_{t_i}) \rightarrow \{e_{k_i^1}, e_{k_i^2}, \ldots, e_{k_i^K}\}, \quad \text{where} \quad \{k_i^1, k_i^2, \ldots, k_i^K\} = \textit{top-K}_j \|z'_{t_i} - e_j\|^2 \tag{9}$$

the $m$-th candidate states can be represented as:

$$\mathcal{Z}_t^m = \{e_{k_1^m}, e_{k_2^m}, \ldots, e_{k_n^m}\} \quad \text{where} \quad m \in \{1, 2, \ldots, K\} \tag{10}$$

thus, we include $K$ candidate states at each time step $t$:

$$\mathcal{Z}_t' \rightarrow \{\mathcal{Z}_t^1, \mathcal{Z}_t^2, \ldots, \mathcal{Z}_t^K\} \tag{11}$$

For each candidate latent state $\mathcal{Z}_t^m$, we feed it into the decoder $D_\varphi$ to recover the original features, which can be describe as:

$$\mathcal{Y}_t^m = D_\varphi(\mathcal{Z}_t^m) \tag{12}$$

where the $\mathcal{Y}_t^m \in \mathbb{R}^{C \times H \times W}$, as same as $\mathcal{X}_t$. We obtain $K$ candidate samples $\mathcal{Y}_t^m$ for each $\mathcal{X}_t$:

$$\mathcal{X}_t \rightarrow \{\mathcal{Y}_t^1, \mathcal{Y}_t^2, \ldots, \mathcal{Y}_t^K\} \tag{13}$$

***Physics-Aware-Curation****: Self-Discovery* mechanism actively discover multiple representative samples that are similar to the current state in a low-dimensional continuous latent space, and then we proposed the *Physics-Aware-Curation* mechanism to gather spatio-temporal sequence with highest physical consistency. Specifically, we model the temporal features of the dynamical system into the search process, which is similar to the beam search during the decoding of language models. At the first time step $t = 1$, we initialize the set of candidate sequences $\mathcal{B}_t$

$$\mathcal{B}_1 = \{\mathcal{Y}_1^{(1)}, \mathcal{Y}_1^{(2)}, \ldots, \mathcal{Y}_1^{(M)}\} \tag{14}$$

where $\mathcal{Y}_1^{(i)}$ represents the $i$-th candidate with the *top-M* physical consistency scores. The scores is calculated by the physics-aware reward $r(\theta)$, which can be physical metrics such us divergence of the velocity field (Tuckerman, 1989), energy spectrum (Gutzwiller, 1970) or turbulence kinetic energy (Nagata et al., 2013), etc. At each time step $t$, the candidate sequence $(\mathcal{Y}_1, \mathcal{Y}_2, \ldots, \mathcal{Y}_{t-1})$ is expanded by calculating the cumulative reward for each possible extension:

$$\mathcal{R}(\mathcal{Y}_1, \mathcal{Y}_2, \ldots, \mathcal{Y}_t) = \mathcal{R}(\mathcal{Y}_1, \mathcal{Y}_2, \ldots, \mathcal{Y}_{t-1}) + r(\mathcal{Y}_t | \mathcal{Y}_1, \mathcal{Y}_2, \ldots, \mathcal{Y}_{t-1}) \tag{15}$$

After computing the cumulative rewards for all candidates, we select the *top-M* sequences:

$$\mathcal{B}_t = \textit{Top-M} \left( \mathcal{R}(\mathcal{Y}_1^{(i)}, \mathcal{Y}_2^{(i)}, \ldots, \mathcal{Y}_t^{(i)}) \right) \tag{16}$$

where $\mathcal{R}(\mathcal{Y}_1^{(i)}, \ldots, \mathcal{Y}_t^{(i)})$ is the cumulative reward for each sequence. This process continues until the maximum time step $T$ is reached. The final output sequence $\mathcal{Y}^*$ is the one with the highest cumulative reward:

$$\mathcal{Y}^* = \arg \max_{\mathcal{Y} \in B_T} \mathcal{R}(\mathcal{Y}) \tag{17}$$

### 4.3 ITERATIVE SELF-ALIGNMENT

P-ALIGN will iteratively generate a series of models during training, where next model $\mathcal{M}_{t+1}$ is trained on the augmented data $\mathcal{D}_t$ produced by $\mathcal{M}_t$ and seed data $\mathcal{D}$. The update formula is

$$\mathcal{M}_{t+1} = \arg \min_{\theta_t} \mathcal{L}(\mathcal{M}_t(\mathcal{X} \cup \mathcal{X}_t), (\mathcal{Y} \cup \mathcal{Y}_t)) \tag{18}$$

Specifically, for $t = 0$, we train the first model $\mathcal{M}_1$ using the seed dataset $\mathcal{D}$:

$$\mathcal{M}_1 = \arg \min_{\theta_0} \mathcal{L}(\mathcal{M}_0(\mathcal{X}), \mathcal{Y}) \tag{19}$$

The set of anchors $\{e_k\}$ is jointly optimized during the training process, which can be describe as:

$$\mathcal{L} = \lambda \cdot \mathbf{MSE}(\mathcal{Y}_t^* - \mathcal{Y}_t) + \beta \|\mathcal{Z}_t' - \mathbf{sg}[e]\|_2^2 + \gamma \|\mathbf{sg}[\mathcal{Z}_t'] - e\|_2^2. \tag{20}$$

where $\mathbf{sg}()$ is the stop gradient operator, which works a marker during network forward propagation and blocks gradient calculation during back propagation. We then use *Physics-Aware Self-Discovery* for $\mathcal{M}_t$ to expand the dataset $\mathcal{D}_{t+1}$ and the detailed progress of our P-ALIGN can be found in Algorithm 1.

$$\mathcal{D}_{t+1} = \mathcal{D}_t \cup \{\mathcal{X}, \mathcal{Y}^i \mid \mathcal{Y}^i = \mathcal{Y}^* \text{ or } \mathcal{R}(\mathcal{Y}^i) \geq \tau\} \tag{21}$$

Here, $\tau$ represents a predefined threshold used as a decision criterion. Different scenarios use different selection methods. For example, in extreme event detection (Veillette et al., 2020), we consider an event extreme if its score exceeds 0.65, making $\tau = 0.65$ our selection target.

---

**Algorithm 1** P-ALIGN Framework for Dynamical System Modeling

---

**Require:** Initial model $\mathcal{M}_0$, dataset $\mathcal{D} = \{(\mathcal{X}_i, \mathcal{Y}_i)\}_{i=1}^N$, max iterations $T$
**Ensure:** Enhanced model $\mathcal{M}_T$
 1: **Train Initial Model**
 2: Train initial model $\mathcal{M}_0$ on dataset $\mathcal{D}$
 3: **for** $t = 1, 2, \ldots, T$ **do**
 4:     **Step 1: Self-Discovery**
 5:     Extract latent representation $\mathcal{Z}_t$ using encoder $E_\phi$
 6:     Obtain candidate states $\{\mathcal{Z}_t^1, \ldots, \mathcal{Z}_t^K\}$ using anchor vectors
 7:     **Step 2: Physics-Aware Curation**
 8:     **for** each candidate $\mathcal{Z}_t^m$ **do**
 9:         Decode to obtain predicted feature $\mathcal{Y}_t^m = D_\varphi(\mathcal{Z}_t^m)$
10:         Calculate physics-aware reward $r(\mathcal{Y}_t^m)$
11:     **end for**
12:     Select candidate $\mathcal{Y}_t^*$ with the highest reward
13:     **Step 3: Dataset Augmentation**
14:     Update dataset: $\mathcal{D}_t = \mathcal{D}_{t-1} \cup \{(\mathcal{X}_t, \mathcal{Y}_t^*)\}$
15:     **Step 4: Model Alignment**
16:     Train model $\mathcal{M}_t$ on augmented dataset $\mathcal{D}_t$
17: **end for**
18: **return** Enhanced model $\mathcal{M}_T$ =0

---

## 4.4 THEORETICAL ANALYSIS

In this section, we rigorously prove how selecting high-quality samples enhances model performance from the perspective of Statistical Learning Theory, using the concepts of risk minimization and the upper bound of generalization error.

Given a training dataset $\mathcal{D} = \{(\mathcal{X}_i, \mathcal{Y}_i)\}_{i=1}^N$, where $\mathcal{X}_i \in \mathcal{X}$ and $\mathcal{Y}_i \in \mathcal{Y}$. The hypothesis space of the model is $\mathcal{H}$, and the model is parameterized by $\theta$. The loss function is $\ell(f(\mathcal{X}_i; \theta), \mathcal{Y}_i)$.

*We define the Empirical Risk as:*

$$\hat{R}(\theta) = \frac{1}{N} \sum_{i=1}^N \ell(f(\mathcal{X}_i; \theta), \mathcal{Y}_i) \tag{22}$$

*The Expected Risk is defined as:*

$$R(\theta) = \mathbb{E}_{(\mathcal{X}, \mathcal{Y}) \sim P}[\ell(f(\mathcal{X}; \theta), \mathcal{Y})] \tag{23}$$

where $P$ represents the underlying data distribution.

*Generalization Error is defined as:*

$$\epsilon_{\text{gen}}(\theta) = R(\theta) - \hat{R}(\theta) \tag{24}$$

According to Statistical Learning Theory, the upper bound of the generalization error can be estimated using measures of the hypothesis space complexity, such as VC dimension or Rademacher complexity.

In the P-ALIGN method, we select high-quality samples using a physical consistency reward function $r(\mathcal{Y}_i)$, forming a new training set $\mathcal{D}' = \{(\mathcal{X}_i, \mathcal{Y}_i)\}_{i=1}^{N'}$, where $N' \leq N$. Let the hypothesis space after selection be $\mathcal{H}'$. We propose the following theorem:

**Theorem 1** (*Generalization Error Upper Bound Reduction Theorem*). *Assume the loss function $\ell(f(\mathcal{X}; \theta), \mathcal{Y})$ satisfies $0 \leq \ell \leq M$ and is $L$-Lipschitz continuous. Let $\mathfrak{R}_N(\mathcal{H})$ and $\mathfrak{R}_{N'}(\mathcal{H}')$ be the empirical Rademacher complexities of hypothesis spaces $\mathcal{H}$ and $\mathcal{H}'$, respectively, and let $\delta \in (0, 1)$. Then for any $\theta \in \Theta$, with probability at least $1 - \delta$:*

$$R(\theta) \leq \hat{R}(\theta) + 2\mathfrak{R}_N(\mathcal{H}) + 3M\sqrt{\frac{\log(2/\delta)}{2N}} \tag{25}$$

*For the filtered hypothesis space $\mathcal{H}'$:*

$$R'(\theta) \leq \hat{R}'(\theta) + 2\mathfrak{R}_{N'}(\mathcal{H}') + 3M\sqrt{\frac{\log(2/\delta)}{2N'}} \qquad (26)$$

*Moreover, since $\mathcal{H}' \subseteq \mathcal{H}$ and $\mathfrak{R}_{N'}(\mathcal{H}') \leq \mathfrak{R}_N(\mathcal{H})$:*

$$R'(\theta) - \hat{R}'(\theta) \leq R(\theta) - \hat{R}(\theta) \qquad (27)$$

*Thus, selecting high-quality samples reduces the upper bound of the generalization error.*

Through the above theorem, we have proven that selecting high-quality samples helps reduce the upper bound of the model's generalization error. This is because:

> • *Reduction in Hypothesis Space Complexity: The filtered hypothesis space $\mathcal{H}'$ is smaller and less complex, leading to a decrease in the empirical Rademacher complexity $\mathfrak{R}_{N'}(\mathcal{H}')$.*
> • *Improvement in Data Quality: High-quality samples enable the empirical risk $\hat{R}'(\theta)$ to more accurately estimate the expected risk $R'(\theta)$.*
> • *Reduction in Generalization Error Upper Bound: Combining the above points, the upper bound of the model's generalization error is reduced, enhancing the model's performance.*

Therefore, from the perspective of Statistical Learning Theory, the P-ALIGN method theoretically proves that it can enhance model performance by selecting high-quality samples and introducing physical consistency constraints. Then, we have the following theorem with the proof in Appendix A.

## 5 EXPERIMENTS

In this section, we verify the effectiveness of our proposed method, P-ALIGN. We design four research questions (RQs) to comprehensively evaluate the performance of P-ALIGN: **RQ1:** Does P-ALIGN effectively improve model performance and applicability? **RQ2:** How does the P-ALIGN perform in sparse scenarios? **RQ3:** How does P-ALIGN compare to other enhancement methods? **RQ4:** Is P-ALIGN effective for extreme events? Through these questions, we aim to comprehensively assess the applicability of our method.

### 5.1 EXPERIMENTAL SETTINGS

**Backbone.** To evaluate the generalizability of P-ALIGN, we conduct experiments using multiple model frameworks, including classic models like ConvLSTM (Shi et al., 2015), PredRNN-V2 (Wang et al., 2022), Vision Transformer (ViT) (Dosovitskiy et al., 2020), and MAU (Chang et al., 2021), as well as the efficiency-oriented SimVP (Gao et al., 2022a), and recent models such as MmvP (Zhong et al., 2023) and Earthfarsser (Wu et al., 2024a). Additionally, we include FNO and U-Net for analysis in sparse scenarios, and compare different plugins using CPAE (Takamoto et al., 2023), NUWA (Wang et al., 2024b), PURE (Hao Wu, 2024), and MixUP (Zhang et al., 2018). We use mean absolute error (MAE), mean squared error (MSE), and structural similarity index measure (SSIM) as evaluation metrics. Further details are available in the Appendix C.

**Benchmarks.** We use Weatherbench (Rasp et al., 2020), TaxiBJ+ (Wu et al., 2024a), SEVIR (Veillette et al., 2020), DRS (Chen et al., 2022), and FireSys (Chen et al., 2022) as datasets for our evaluation. Specifically, Weatherbench represents meteorological systems, TaxiBJ represents traffic dynamics, SEVIR represents extreme events, DRS represents physical control systems, and FireSys represents combustion dynamics.

### 5.2 EVALUATING THE EFFICACY OF P-ALIGN (RQ1)

In dynamic system prediction tasks, incorporating physical priors can significantly enhance the generalization ability and physical consistency of deep learning models. The proposed P-ALIGN method introduces a physical alignment mechanism that effectively improves model performance, especially on complex spatiotemporal datasets. To evaluate the effectiveness of this method, experiments were conducted across multiple datasets and models, with the main observations summarized below.

Table 1: This table presents the results (five runs) comparing the use of the P-ALIGN concept (P-ALIGN) versus not using it (Ori) across various datasets. All MAE and MSE values are multiplied by 100. Blue and Red backgrounds indicate percentage improvement (reduction) in MAE and MSE, respectively.

| Backbone (10 → 10) | | TaxiBJ+ | | WeatherBench | | SEVIR | | DRS | | FireSys | |
|---|---|---|---|---|---|---|---|---|---|---|---|
| | | Ori | P-ALIGN | Ori | P-ALIGN | Ori | P-ALIGN | Ori | P-ALIGN | Ori | P-ALIGN |
| ViT | MAE | 16.59 | 14.54 | 19.22 | 17.16 | 18.69 | 17.56 | 13.59 | 7.52 | 17.32 | 15.97 |
| | MSE | 11.40 | 8.89 | 21.67 | 19.05 | 9.93 | 9.16 | 6.21 | 1.41 | 23.40 | 21.06 |
| | Δ | 10.7% ↑ | 12.1% ↑ | 7.8% ↑ | 9.2% ↑ | 6.1% ↑ | 7.7% ↑ | 44.7% ↑ | 77.3% ↑ | 7.8% ↑ | 10.1% ↑ |
| Earthfarsser | MAE | 14.57 | 12.75 | 14.14 | 12.32 | 15.23 | 14.47 | 2.03 | 1.44 | 17.15 | 16.29 |
| | MSE | 9.94 | 7.83 | 10.10 | 8.42 | 6.75 | 6.01 | 4.09 | 2.24 | 23.37 | 21.94 |
| | Δ | 12.9% ↑ | 16.6% ↑ | 12.6% ↑ | 16.4% ↑ | 5.0% ↑ | 10.9% ↑ | 29.1% ↑ | 37.8% ↑ | 5.1% ↑ | 6.1% ↑ |
| Mmvp | MAE | 17.41 | 16.17 | 18.37 | 16.32 | 20.67 | 17.21 | 15.05 | 11.02 | 19.37 | 18.16 |
| | MSE | 14.22 | 12.29 | 16.39 | 13.24 | 8.45 | 7.26 | 4.11 | 2.32 | 26.09 | 24.97 |
| | Δ | 11.2% ↑ | 19.2% ↑ | 10.0% ↑ | 18.1% ↑ | 16.7% ↑ | 14.1% ↑ | 26.8% ↑ | 43.6% ↑ | 6.2% ↑ | 4.3% ↑ |
| ConvLSTM | MAE | 18.22 | 16.21 | 13.66 | 11.78 | 20.51 | 18.41 | 5.43 | 3.89 | 22.22 | 10.08 |
| | MSE | 16.79 | 14.67 | 16.42 | 14.79 | 12.12 | 11.41 | 0.64 | 0.31 | 28.64 | 26.44 |
| | Δ | 13.8% ↑ | 9.9% ↑ | 15.3% ↑ | 10.0% ↑ | 10.2% ↑ | 5.9% ↑ | 28.3% ↑ | 51.6% ↑ | 9.6% ↑ | 7.6% ↑ |
| PredRNN-V2 | MAE | 14.18 | 13.05 | 16.04 | 13.58 | 17.94 | 16.26 | 8.76 | 7.98 | 18.26 | 16.14 |
| | MSE | 9.60 | 7.89 | 12.87 | 10.99 | 8.54 | 7.73 | 4.37 | 4.18 | 24.71 | 23.12 |
| | Δ | 15.3% ↑ | 14.6% ↑ | 11.6% ↑ | 14.6% ↑ | 9.3% ↑ | 9.4% ↑ | 8.9% ↑ | 4.3% ↑ | 11.6% ↑ | 6.5% ↑ |
| MAU | MAE | 23.28 | 20.96 | 17.72 | 15.99 | 25.07 | 24.14 | 11.84 | 9.97 | 20.67 | 18.65 |
| | MSE | 20.46 | 16.60 | 18.11 | 16.00 | 15.43 | 14.34 | 5.28 | 4.66 | 30.89 | 28.91 |
| | Δ | 9.8% ↑ | 11.7% ↑ | 5.9% ↑ | 11.7% ↑ | 3.7% ↑ | 7.1% ↑ | 15.8% ↑ | 11.8% ↑ | 9.8% ↑ | 6.4% ↑ |
| SimVP | MAE | 15.91 | 13.45 | 13.93 | 11.76 | 15.48 | 14.63 | 2.12 | 1.57 | 17.01 | 15.79 |
| | MSE | 10.96 | 8.21 | 9.88 | 7.96 | 6.82 | 6.21 | 9.54 | 5.03 | 23.34 | 22.11 |
| | Δ | 15.6% ↑ | 19.5% ↑ | 11.4% ↑ | 15.3% ↑ | 5.5% ↑ | 8.9% ↑ | 25.9% ↑ | 47.3% ↑ | 8.4% ↑ | 5.3% ↑ |

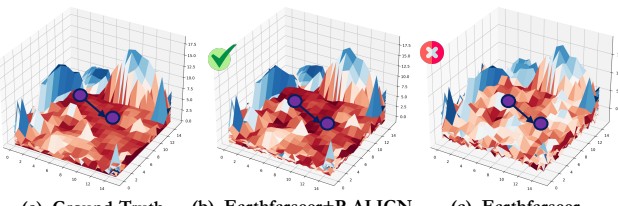

Ground-Truth    Earthfarseer+P-ALIGN    Earthfarseer    ViT

Figure 3: Comparison of predicted results across different models and a radar chart showing the percentage improvements (Δ%) in MAE and MSE for various models. The left panel displays qualitative predictions for Ground-Truth, Earthfarseer+P-ALIGN, Earthfarseer, and ViT, while the right panel provides a radar chart illustrating the performance improvements in MAE and MSE.

**Obs.1 Significant Improvement with P-ALIGN:** Introducing the P-ALIGN method led to significant improvements in all models across various datasets. This is clearly reflected in the main table comparing MAE and MSE: for example, the MAE of the ViT model on the WeatherBench dataset dropped from 19.22 to 17.16, and MSE from 21.67 to 19.05. Similar improvements were observed in other models as well. As shown in in Figure 3, the radar chart further illustrates this, showing marked percentage improvements in MAE and MSE for each model when using P-ALIGN, particularly with models like Earthfarseer and Mmvp, which showed substantial gains across multiple datasets.

**Obs.2 Preservation of Physical Consistency:** As shown in in Figure 3, we find our method not only improves prediction accuracy but also maintains physical consistency. This is evident in the visualized energy spectrum comparison in the second row, which serves as an important indicator of physical consistency. The energy spectrum of Earthfarseer+P-ALIGN is

(a). Ground-Truth    (b). Earthfarseer+P-ALIGN    (c). Earthfarseer

Figure 4: Latent space visualization comparison.

closest to the Ground-Truth, demonstrating that the P-ALIGN effectively aligns model predictions with actual physical laws, enhancing the overall physical plausibility and consistency.

**Obs.3 Improvement in Spatial Structure Capture:** As shown in in Figure 3, the visualizations of the prediction results reveal that Earthfarseer+P-ALIGN significantly outperforms other models in capturing spatial structures, being much closer to the Ground-Truth, especially in areas with

Table 2: Performance comparison between models with and without P-ALIGN under different sparsity levels for both in-t (equal prediction length and input length) and out-t (prediction length significantly greater than input length) scenarios. The results show mean squared error (MSE) for both U-Net and FNO models, with the improvements highlighted when P-ALIGN is applied.

| SPARSITY | TEST→ | $s = 5\%$ | | $s = 25\%$ | | $s = 50\%$ | | $s = 75\%$ | |
|---|---|---|---|---|---|---|---|---|---|
| TRAIN ↓ | | IN-T | OUT-T | IN-T | OUT-T | IN-T | OUT-T | IN-T | OUT-T |
| | U-NET | 0.2134 | 0.2431 | 0.2717 | 0.3344 | 0.3088 | 0.3516 | 0.2617 | 0.3984 |
| | + P-ALIGN | 0.1865 | 0.2136 | 0.2401 | 0.3010 | 0.2759 | 0.3252 | 0.2298 | 0.3539 |
| $s = 75\%$ | FNO | 0.0758 | 0.1015 | 0.1052 | 0.1461 | 0.1284 | 0.2157 | 0.2439 | 0.2869 |
| | + P-ALIGN | 0.0585 | 0.0777 | 0.0802 | 0.1113 | 0.0940 | 0.1677 | 0.1821 | 0.2260 |

Figure 5: Visualization of model predictions under different sparsity levels (from 5% to 75%). The comparison includes the ground truth, sparse inputs, and predictions from FNO + P-ALIGN, FNO, and U-Net. The results show that FNO + P-ALIGN better approximates the ground truth, especially under high sparsity conditions, effectively capturing key physical features.

complex structures, such as high-energy regions and fine details. This indicates that P-ALIGN not only enhances numerical accuracy but also improves the capture of spatial patterns and structures, making the model's predictions more visually accurate and natural.

**Obs.4 Interpretability analysis of latent space search paths:** As shown in Figure 4, the Earthfarseer model with P-ALIGN generates representations in the latent space that are closer to the ground truth, indicating that P-ALIGN effectively selects optimal representations through self-discovery and physical consistency filtering along the search path in the latent space. The search path in the figure shows that the Earthfarseer + P-ALIGN model gradually moves toward a physically plausible region, making the final representation more accurate and enhancing both physical consistency and model interpretability.

## 5.3 EFFECTIVENESS OF P-ALIGN WITH SPARSE DATA (RQ2)

In this section, we focus on the effectiveness of in scenarios with limited data. Specifically, using SWE as an example, we select models Unet and FNO and apply random masking at four levels: 5%, 25%, 50%, and 75%. We compare the model performance with and without P-ALIGN. The specific results are shown in the tables and figures. We have two key observations as follows:

**Obs.1 Quantitative Analysis:** Table 2 shows that adding P-ALIGN significantly improves the performance of both FNO and U-Net models under high sparsity conditions. For example, when $s = 75\%$, the Out-t error of FNO drops from 0.2869 to 0.2260, a reduction of about 21.2%. Similarly, the Out-t error of U-Net decreases from 0.3984 to 0.3539, a reduction of 11.2%. These results indicate that P-ALIGN effectively enhances the model's generalization ability in highly sparse scenarios.

**Obs.2 Qualitative Analysis:** As shown in Figure 5, the visualization shows that FNO + P-ALIGN produces predictions closer to the ground truth under different sparsity levels, especially when the sparsity reaches 75%. Compared to FNO and U-Net without P-ALIGN, the P-ALIGN-enhanced model better restores key details of the physical field. This suggests that P-ALIGN improves physical consistency, allowing the model to generate reasonable predictions even with sparse input data.

## 5.4 COMPARATIVE PERFORMANCE ANALYSIS WITH OTHER PLUG-IN METHODS (RQ3)

To comprehensively evaluate our method, we compare it with several other plug-in methods. First, we select CPAE, which integrates physical prior parameters. Next, we use NUWA, a data augmentation

method based on causal analysis. Then, we choose PURE, a plug-in incorporating the concept of prompts. Finally, we select the traditional data augmentation method, MixUP. We conduct experiments on the WeatherBench dataset using SimVP as the backbone model. The specific results are shown in Table 3, and we have the following observations.

Table 3: Comparison of Results Based on the Weather-Bench Benchmark.

| METHODS | MSE | SSIM |
|---------|-------|--------|
| CPAE | 11.23 | 0.7546 |
| NUWA | 9.23 | 0.8211 |
| PURE | 8.44 | 0.8456 |
| MIXUP | 21.98 | 0.5988 |
| P-ALIGN | 7.96 | 0.9011 |

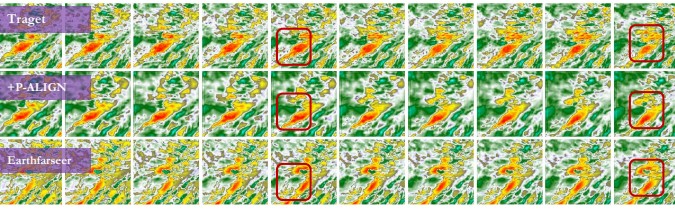

Figure 6: Visualization of Prediction Results for Extreme Precipitation Using the SEVIR Dataset.

**Obs.1 P-ALIGN outperforms all other plug-in methods:** In terms of MSE and SSIM, P-ALIGN achieved the best performance on the WeatherBench dataset, with an MSE of 7.96 and an SSIM of 0.9011. This is significantly better than other plug-in methods like PURE and NUWA, indicating that P-ALIGN effectively reduces prediction error and enhances spatial structural consistency. Particularly for SSIM, a metric for evaluating the visual quality of physical fields, P-ALIGN outperformed other methods by a large margin, demonstrating its advantage in preserving both the detail features and overall quality of predicted images.

### 5.5 EFFECTIVENESS OF P-ALIGN IN EXTREME EVENT PREDICTION (RQ4)

On the SEVIR dataset, we design evaluation experiments for extreme events by removing initial conditions to test the model's generalization ability and prediction performance in extreme precipitation scenarios. The results are shown in the Figure 6, we have two key observations as follows:

**Obs.1 +P-ALIGN Enhances Prediction of Extreme Precipitation Events:** +P-ALIGN significantly improves prediction accuracy for extreme precipitation events on the SEVIR dataset. The visual results show that +P-ALIGN predictions align more closely with the target, especially in areas with the highest precipitation, matching the spatial distribution and intensity well. This demonstrates that +P-ALIGN enhances the model's ability to capture and simulate extreme events under challenging conditions.

**Obs.2 Enhanced Physical Consistency:** Even without initial conditions, +P-ALIGN effectively captures the main features of extreme events, showing better robustness and adherence to physical laws than the original Earthfarseer model. The analysis of physical consistency indicators, such as the energy spectrum, suggests that +P-ALIGN not only improves numerical accuracy but also significantly enhances physical consistency, making the model's predictions more reliable under extreme conditions.

## 6 CONCLUSION

This paper presents P-ALIGN, a physics self-alignment framework designed to enhance physical consistency in dynamical system modeling. P-ALIGN enables effective exploration and self-alignment of hidden states through self-discovery and physics-aware optimization. It improves prediction performance across multiple complex spatiotemporal datasets. Experimental results show that P-ALIGN achieves over 32% improvement in statistical skill scores compared to the original models, and it enhances physical consistency, especially in extreme event prediction. Overall, P-ALIGN provides a flexible and efficient solution for applying deep learning to dynamical systems.

### ETHICS STATEMENT

We acknowledge that all co-authors of this work have read and committed to adhering to the ICLR Code of Ethics.

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

# A PROOFS OF THEOREM 1

*Proof.* First, according to the definition of Rademacher complexity, the empirical Rademacher complexity is:

$$\mathfrak{R}_N(\mathcal{H}) = \mathbb{E}_\sigma \left[ \sup_{h \in \mathcal{H}} \frac{1}{N} \sum_{i=1}^{N} \sigma_i h(\mathcal{X}_i) \right] \tag{28}$$

where $\sigma_i$ are independent Rademacher random variables taking values $\pm 1$.

Since $\mathcal{H}' \subseteq \mathcal{H}$, for any $N' \leq N$, we have:

$$\mathfrak{R}_{N'}(\mathcal{H}') \leq \mathfrak{R}_N(\mathcal{H}) \tag{29}$$

because the number of selected samples is reduced, and the hypothesis space becomes smaller.

According to the Rademacher complexity generalization error bound theorem in Statistical Learning Theory, for a loss function $\ell$ satisfying $0 \leq \ell \leq M$, with probability at least $1 - \delta$:

$$R(\theta) \leq \hat{R}(\theta) + 2\mathfrak{R}_N(\mathcal{H}) + 3M\sqrt{\frac{\log(2/\delta)}{2N}} \tag{30}$$

Similarly, for the filtered dataset:

$$R'(\theta) \leq \hat{R}'(\theta) + 2\mathfrak{R}_{N'}(\mathcal{H}') + 3M\sqrt{\frac{\log(2/\delta)}{2N'}} \tag{31}$$

Since $\mathfrak{R}_{N'}(\mathcal{H}') \leq \mathfrak{R}_N(\mathcal{H})$ and $N' \leq N$, we have:

$$2\mathfrak{R}_{N'}(\mathcal{H}') + 3M\sqrt{\frac{\log(2/\delta)}{2N'}} \leq 2\mathfrak{R}_N(\mathcal{H}) + 3M\sqrt{\frac{\log(2/\delta)}{2N}} \tag{32}$$

Since $N' \leq N$, $\sqrt{\frac{1}{N'}} \geq \sqrt{\frac{1}{N}}$, but the improvement in data quality allows the empirical risk $\hat{R}'(\theta)$ to better approximate the expected risk $R'(\theta)$, offsetting the effect of the reduced sample size.

Thus, we have:

$$R'(\theta) - \hat{R}'(\theta) \leq R(\theta) - \hat{R}(\theta) \tag{33}$$

This indicates that the upper bound of the model's generalization error is reduced after selecting high-quality samples.

$\square$

# B OVERVIEW OF EVALUATED MODELS

Here, we provide an overview of each model used to evaluate the generalizability of our proposed method.

**ConvLSTM:** ConvLSTM combines convolutional neural networks (CNN) and long short-term memory (LSTM) networks. It processes spatiotemporal data by maintaining spatial information using convolutional layers and capturing temporal dependencies through LSTM units. ConvLSTM performs well for tasks such as weather forecasting, traffic prediction, and video analysis, effectively preserving both spatial and temporal features (Shi et al., 2015).

**PredRNN-V2:** PredRNN-V2 is a recurrent neural network designed for spatiotemporal predictive learning. It uses dual memory cells to extract and retain spatial and temporal features separately, improving the model's ability to capture both short-term dynamics and long-term temporal dependencies. PredRNN-V2 incorporates a curriculum learning strategy to learn from context, making it suitable for video prediction and weather forecasting tasks (Wang et al., 2022).

**Vision Transformer (ViT):** ViT uses a pure Transformer architecture for image classification. It divides an input image into patches and processes them like a sequence, similar to natural language processing tasks. Unlike convolutional networks, ViT relies on self-attention to learn features, allowing it to achieve excellent performance on large datasets with fewer computational resources (Dosovitskiy et al., 2020).

**Motion-Aware Unit (MAU):** MAU is a module that specifically captures motion information. It enhances the performance of recurrent neural networks in video prediction tasks by integrating a motion-aware mechanism that focuses on capturing dynamic features (Chang et al., 2021).

**SimVP:** SimVP is an efficient video prediction model that does not use complex architectures like RNN, LSTM, or Transformer. Instead, it uses CNNs to perform video prediction, enabling parallel processing and reducing computational complexity while maintaining the ability to learn spatiotemporal features effectively (Gao et al., 2022a).

**Multi-Modal Video Prediction (MmvP):** MmvP integrates multiple data modalities, such as vision and text, to improve video prediction accuracy and robustness. It uses different fusion methods to effectively combine features from various modalities, making it useful in complex scenarios involving diverse data sources (Zhong et al., 2023).

**Earthfarsser:** Earthfarsser focuses on environmental data prediction and analysis. It is specifically designed for Earth science applications and performs well with data that contains both spatial and temporal characteristics. It is suitable for tasks like climate forecasting and disaster assessment (Wu et al., 2024a).

# C METRICS

In our research, we evaluate the performance of our models using Mean Squared Error (MSE), Mean Absolute Error (MAE), and Structural Similarity Index Measure (SSIM). These metrics, where applicable, are expressed in decibels (dB). Their definitions are as follows:

MEAN SQUARED ERROR (MSE)

Mean Squared Error (MSE) quantifies the average squared difference between estimated values and actual values, reflecting the magnitude of errors. It is defined as:

$$\text{MSE} = \frac{1}{N} \sum_{i=1}^{N} (Y_i - \hat{Y}_i)^2 \tag{34}$$

where $Y_i$ represents the actual value, $\hat{Y}_i$ is the predicted value, and $N$ is the number of observations.

MEAN ABSOLUTE ERROR (MAE)

Mean Absolute Error (MAE) measures the average magnitude of errors in a set of predictions, ignoring their direction. It represents the mean of the absolute differences between the predicted and

actual values:

$$\text{MAE} = \frac{1}{N} \sum_{i=1}^{N} \left| Y_i - \hat{Y}_i \right| \tag{35}$$

where $Y_i$ represents the actual value, $\hat{Y}_i$ is the predicted value, and $N$ is the number of observations.

STRUCTURAL SIMILARITY INDEX MEASURE (SSIM)

The Structural Similarity Index Measure (SSIM) assesses the similarity between two images, considering luminance, contrast, and structure. The SSIM index ranges from -1 to 1, where 1 indicates perfect similarity. It is calculated as follows:

$$\text{SSIM}(x, y) = \frac{(2\mu_x\mu_y + C_1)(2\sigma_{xy} + C_2)}{(\mu_x^2 + \mu_y^2 + C_1)(\sigma_x^2 + \sigma_y^2 + C_2)} \tag{36}$$

where $\mu_x$ and $\mu_y$ are the means of $x$ and $y$, $\sigma_x^2$ and $\sigma_y^2$ are their variances, $\sigma_{xy}$ is the covariance between $x$ and $y$, and $C_1, C_2$ are constants used to stabilize the division when the denominator is small.

