# OpenReview forum: "P-Align: Self-Alignment in Physical Dynamical System Modeling"
_ICLR.cc/2025/Conference — ICLR 2025 Conference Withdrawn Submission_

### Official Review · Reviewer_BwBS · 2024-10-18

**Soundness:** 2
**Presentation:** 3
**Contribution:** 3
**Rating:** 6
**Confidence:** 3

**Summary:**

This paper introduces a self-alignment framework to improve the physics consistency in dynamics modeling. The framework works by iteratively augmenting the training dataset with samples that align more closely with predefined physics-aware metrics, and then refining the model using the augmented dataset. The proposed approach is validated across various backbone networks and compared with other enhancement methods.

**Strengths:**

This paper is well-structured, providing a thorough introduction to the background and limitations of related work, which effectively clarifies the motivation behind this study. The well-designed figures aid in understanding the framework of the proposed method. Furthermore, the method is validated across a broad range of backbone networks, demonstrating its strong generalization capability.

**Weaknesses:**

1.  While the method is well-motivated, a more in-depth analysis is needed to explain why improving physics consistency of the training samples leads to better prediction performance.
- **A**. Although the authors mention several metrics that can serve as physics-aware rewards, they lack implementation details, such as which metrics are applied to each dataset and how they are computed.
- **B**. A quantitative and qualitative comparison between the original data and the augmented data in terms of physics-aware metrics is necessary to demonstrate the effectiveness of the self-alignment process.
- **C**. Since the paper lacks training details (the number of alignment iterations and the training steps in each iteration), there remains a possibility that the performance improvements result from increased training schedule. To better understand the impact of physics-consistent training samples, two additional experiments should be provided: (i) a comparison with baselines trained under the same computational budget, and (ii) training models from scratch with samples at varying levels of physics consistency, which can be obtained from previous runs of P-Align and categorized based on the physics-aware metrics.

2. While P-Align is compared with other plug-in enhancement methods, none of these methods utilize the physics-aware metrics. To provide a more comprehensive analysis, it would be beneficial to compare P-Align with approaches that incorporate such metrics directly into their loss functions. Although these methods may face optimization challenges, as suggested in the Introduction (Line 53), experimental evidence is necessary to substantiate this claim.

**Questions:**

See Weaknesses.

---

### Official Review · Reviewer_mwn9 · 2024-10-27

**Soundness:** 2
**Presentation:** 2
**Contribution:** 2
**Rating:** 3
**Confidence:** 4

**Summary:**

This paper aims to address the problem of physical dynamical systems.  It proposes a framework, named P-Align, including four modules: (1) self-discovery module to discretize the hidden representation; (2) physics-aware-curation module to filter samples with higher rewards; (3) data augmentation module to enrich the training set; (4) alignment module to evolve the model. The experimental results on  five benchmarks suggest that P-Align can achieve the best precision (MAE and MSE), and the predicted dynamics perform better   consistency.

**Strengths:**

The experiments are comprehensive. P-Align uses various backbone models and can serve as a plug-in to align the energy spectrum better. Moreover, the paper conducts many analyses, such as in sparse cases, and quantitative analysis.

**Weaknesses:**

1. There are lots of typos in paper:
  + Line 18, the usage of "to provides" needs to be "to provide"
  + Line 26,  Dynamical systems "provides" -> Dynamical systems "provide"
  + In the part of framework overview,  "identify" -> "identifies", "find" -> "finds", "add" -> "adds", "use" -> "uses"
  + Line 168, "a embedding space" -> "an embedding space"
  + Line 180,  "sample" -> "samples", $E\phi$ -> $E_\phi$
  + Line 184, "we assumes" -> "we assume"
  + Line 235, "scores is" -> "scores are", "such us" -> "such as"
  + Line 269, "making ... our selection target" -> "making ... as our selection target"

These are very low-level mistakes, which should not occur in a good paper.

2. In line 170,  $\mathcal{X}_t\in\mathbb{R}^{C \times H\times W}$, you need to introduce the meaning of the three dimensions. Does it stand for an image of the input? If so, what is the physical consistency and priors of image dynamics? I think you selected datasets are not applicable to your model. Those datasets, such as fluid modeled with graphs, may be better.
3. There are no ablation studies to verify the effectiveness of the proposed four modules.
4. The conclusion that better training data have lower generalization error upper bound is trivial. The proofs in the paper are unnecessary.

**Questions:**

1. Are the reward functions $r(\theta)$ pre-trained ?   If so, how do you train it? Because the training set is dynamically increasing, I wonder if the reward functions effective enough for the new training data.
2. From eq. (16), it seems that the indexes of each timestep are the same ( i.e., all $(i)$). If so, how do you do beam search? The indexes of beam search are not restrained to be the same.
3. In eq. (20),  what does $e$ stand for?

---

### Official Review · Reviewer_BMrU · 2024-10-28

**Soundness:** 3
**Presentation:** 1
**Contribution:** 2
**Rating:** 5
**Confidence:** 4

**Summary:**

This paper introduces P-ALIGN, a framework that applies self-alignment concepts from large language models to physical dynamical system modeling. The key innovation is enabling dynamical system models to iteratively improve through self-discovery and physics-aware curation of training data. The framework aims to enhance both statistical accuracy and physical consistency of predictions and is validated by empirical results.

**Strengths:**

1.The design and adaptation of self-alignment concept from LLM to physics system is novel.

2. the designed framework is architecture agnostic and can be used in many scenarios.

3. the paper explains the reduced generalization error upper bound

4. There shows an impressive and consistent improvement across different scenarios

**Weaknesses:**

1.	Many of the implementation details are not clear or missing, see my question 1,2,3

2.	The method requires prior knowledge of physics constraints, which is not necessarily available in real-world scenarios. Also, this could be an unfair comparison with vanilla baseline, since the latter could be easily improved by adding a regularization term to penalize the physics violation.

3.	My major concern is the problem setting in this paper. For dynamical/physics system forecasting specially with these PDE like inputs, using multiple past observation to predict the future is a commonly used and very effective way to improve the forecasting. While this paper seems not using these basics from dynamical systems, therefore the experiments could be less legit. see my question 4.

**Questions:**

1.	Can you clarify the steps in self-discovery section (page 4): how the sub regions/representative vectors are calculated? equation 6 and 8 seems conflicting, is Z’_t calculated by matrix operation or projection to nearest anchor point?

2.	Can you show how the physics rewards function is defined for these specific examples?

3.	Can you show how the detailed structure of these encoder/decoders?

4.	Can you elaborate on problem formulation? Are you mapping from a single step observation to a trajectory (X_t->X_{t+1},X_{t+2..})? If this is the case, the setting is questionable. In practice, a single step might not contain all the information needed to predict the future (i.e. there could exist high-order temporal effect such as acceleration). And in real-world the past observations are available, it is almost free lunch to use that for massively improving the forecasting results. See the section 2.1 in this paper (https://proceedings.neurips.cc/paper_files/paper/2022/hash/87f476af4053961667c2c08e9f4b850e-Abstract-Conference.html) for the standard problem setting in dynamical system learning.

5.	In line 337, the author claims ‘The filtered hypothesis space H′ is smaller’. Can you clarify why it is smaller and smaller compared to which baseline?

---

### Official Review · Reviewer_1CvH · 2024-11-03

**Soundness:** 3
**Presentation:** 1
**Contribution:** 2
**Rating:** 3
**Confidence:** 3

**Summary:**

In this paper, the authors propose a novel framework, Physics-aware Self-Alignment (P-ALIGN), to enhance the physical consistency of dynamical systems modeling. P-ALIGN enables dynamical system models to provides physics-aware rewards, which makes self-alignment of dynamical system models possible. Experimental results show that P-ALIGN achieves over 32% improvement in statistical skill scores compared to the original models.

**Strengths:**

1. The authors innovatively transfer the concept of self-alignment to the field of physical dynamics, providing a new training framework.

2. The authors conducted experiments on various datasets, demonstrating significant performance improvements by combining the P-Align method presented in the paper with various baseline methods.

Originality, quality, clarity, and significance:

The paper is original, and the authors are the first to apply the concept of self-alignment to the field of physical dynamics, opening new avenues for research in this area. The clarity of the paper needs further improvement, and the overall quality is adequate. This research makes a contribution to how more robust models can be trained in the field of physical dynamics based on existing data.

**Weaknesses:**

1. The authors' writing needs further improvement. There are many typos, such as "provides" in line 26, "anchor" in line 182, and "of in" in line 468. Additionally, the authors should reorganize their notation, especially in the "self-discovery" of Section 4.2, where the explanation of Z^{m}_{t} is difficult to understand. Furthermore, the descriptions of equations 6 and 11 seem contradictory.

2. In the authors' framework, the choice of physical metrics appears to play a significant role. Why did the authors select these metrics? Have they considered choosing other metrics?

3. Equations 25 and 26 do not seem to rigorously derive the conclusion of equation 27. Can the authors provide a more rigorous proof?

4. I noticed that the results for the TaxiBJ+ and SEVIR datasets in Table 1 do not match those reported in the original Earthfarsser paper. Can the authors explain the reason for this discrepancy?

5. The baseline methods compared by the authors do not seem to include the methods proposed in the original paper for DRS and FireSys. In this paper's baselines, the newer method MmvP, which is used for macro-motion trajectory prediction, appears unsuitable for physical scenarios like DRS and FireSys. Have the authors considered validating some more recent methods for modeling physical dynamics, such as HOPE [1] and PGODE [2]?

[1] Hope: High-order graph ode for modeling interacting dynamics, ICML 2023.

[2] PGODE: Towards High-quality System Dynamics Modeling, ICML 2024.

**Questions:**

1. How is the specific value of \tau determined in equation 21?

2. The authors mention "removing initial conditions" in lines 511-512. How is this implemented? Is it a form of data preprocessing?

---

### Note · Authors · 2024-11-17

I have read and agree with the venue's withdrawal policy on behalf of myself and my co-authors.